# Constrained Reinforcement Learning Has Zero Duality Gap

**Santiago Paternain, Luiz F. O. Chamon, Miguel Calvo-Fullana and Alejandro Ribeiro**
Electrical and Systems Engineering
University of Pennsylvania
{spater,luizf,cfullana,aribeiro}@seas.upenn.edu

## Abstract

Autonomous agents must often deal with conflicting requirements, such as completing tasks using the least amount of time/energy, learning multiple tasks, or dealing with multiple opponents. In the context of reinforcement learning (RL), these problems are addressed by (i) designing a reward function that simultaneously describes all requirements or (ii) combining modular value functions that encode them individually. Though effective, these methods have critical downsides. Designing good reward functions that balance different objectives is challenging, especially as the number of objectives grows. Moreover, implicit interference between goals may lead to performance plateaus as they compete for resources, particularly when training on-policy. Similarly, selecting parameters to combine value functions is at least as hard as designing an all-encompassing reward, given that the effect of their values on the overall policy is not straightforward. The later is generally addressed by formulating the conflicting requirements as a constrained RL problem and solved using Primal-Dual methods. These algorithms are in general not guaranteed to converge to the optimal solution since the problem is not convex. This work provides theoretical support to these approaches by establishing that despite its non-convexity, this problem has zero duality gap, i.e., it can be solved exactly in the dual domain, where it becomes convex. Finally, we show this result basically holds if the policy is described by a good parametrization (e.g., neural networks) and we connect this result with primal-dual algorithms present in the literature and we establish the convergence to the optimal solution.

## 1 Introduction

Autonomous agents must often deal with conflicting requirements, such as completing a task in the least amount of time/energy, learning multiple tasks or contexts, dealing with multiple opponents or with several specifications that are designed to guide the agent in the learning process. In the context of reinforcement learning [1], these problems are generally addressed by combining modular value functions that encode them individually, by multiplying each signal by its own coefficient, which controls the emphasis placed on it [2–4]. Although effective, the multi-objective problem [5] has several downsides. First, for each set of penalty coefficients, there exists a different, optimal solution, also known as Pareto optimality [6]. In practice, the exact coefficient is selected through a time consuming and a computationally intensive process of hyper-parameter tuning that often times are domain dependent, as showed in [7–9]. Moreover, implicit interference between the goals may lead to training plateaus as they compete for resources in the policy [10].

An alternative, is to embed all conflicting requirements in a constrained RL problem and to use a primal-dual algorithm as in [7, 11] that chooses the parameters automatically. The main advantage of this approach is that constraints ensure satisfying behavior without the need for manually selecting

the penalty coefficients. In these algorithms the policy update is on a faster time-scale than the multiplier update. Thus, effectively, these approaches work as if the dual problem of the constrained reinforcement learning problem was being solved. Thus, guaranteeing to obtain the feasible solution with the smallest suboptimality. Yet, there is no guarantee on how small the suboptimality is. In this work we provide an answer to the previous question. In particular we establish that:

1. Despite its non-convexity, constrained reinforcement learning for policies belonging to a general distribution class has zero duality gap, i.e., it can be solved exactly in the dual domain, where the problem is actually convex

2. Since working with generic distributions as policies is in general intractable, we extend this result to parametrized policies, by showing that the suboptimality bound also holds when the parametrization is a universal approximator, e.g., a neural network [12]).

3. We leverage these theoretical results to establish that the family of primal-dual algorithms for constrained reinforcement learning, e.g. [7, 11], in fact converge to the optimal solution under mild assumptions.

## 1.1 Related Work

Constrained Markov Decision Processes (CMDPs) [13] are an active field of research. CMDP applications cover a vast number of topics, such as: electric grids [14], networking [15], robotics [3, 16, 17] and finance [18, 19]. The most common approaches to solve this problems can be divided under the following categories. **Manual selection of Lagrange multipliers:** constrained Reinforcement Learning problems can be solved through by maximizing an unconstrained Lagrangian, for a specific multiplier [2]. The combination of different rewards with manually selected Lagrange multipliers has been applied for instance to learning complex movements for humanoids [4] or to limit the variance of the constraint that needs to be satisfied [19, 20]. **Integrating prior knowledge** about the system transitions is exploited in order to project the action chosen by the policy to a set that ensures the satisfaction of the constraints [21]. **Primal-dual algorithms** [7, 11], allow us to choose dynamically the multipliers by find the best policy for the current set of parameters and then taking steps along the gradient of the Lagrangian with respect to the multipliers. These allow to consider general constraints and the algorithm is reward agnostic and it does not require the use of prior knowledge.

## 2 Constrained Reinforcement Learning

Let $t \in \mathbb{N} \cup \{0\}$ denote the time instant and $\mathcal{S} \subset \mathbb{R}^n$ and $\mathcal{A} \subset \mathbb{R}^d$ be compact sets describing the possible states and actions of an agent described by a Markovian dynamical system with transition probability density $p$, i.e., $p\left(s_{t+1} \mid \{s_u, a_u\}_{u \leq t}\right) = p\left(s_{t+1} \mid s_t, a_t\right)$ for $s_t \in \mathcal{S}$ and $a_t \in \mathcal{A}$ for all $t$. The agent chooses actions sequentially based on a policy $\pi \in \mathcal{P}(\mathcal{S})$, where $\mathcal{P}(\mathcal{S})$ is the space of probability measures on $(\mathcal{A}, \mathcal{B}(\mathcal{A}))$ parametrized by elements of $\mathcal{S}$, where $\mathcal{B}(\mathcal{A})$ are the Borel sets of $\mathcal{A}$. The action taken by the agent at each state results in rewards defined by the functions $r_i : \mathcal{S} \times \mathcal{A} \to \mathbb{R}$, for $i = 0, \ldots, m$, that the agent accumulates over time. These rewards describe different objectives that the agent must achieve, such as completing a task, remaining within a region of the state space, or not running out of battery. The goal of constrained RL is then to find a policy $\pi^\star \in \mathcal{P}(\mathcal{S})$ that meets these objectives by solving the problem

$$
\begin{aligned}
P^\star \triangleq \max_{\pi \in \mathcal{P}(\mathcal{S})} \quad & V_0(\pi) \triangleq \mathbb{E}_{s,\pi}\left[\sum_{t=0}^{\infty} \gamma^t r_0(s_t, \pi(s_t))\right] \\
\text{subject to} \quad & V_i(\pi) \triangleq \mathbb{E}_{s,\pi}\left[\sum_{t=0}^{\infty} \gamma^t r_i(s_t, \pi(s_t))\right] \geq c_i, i = 1, \ldots, m,
\end{aligned}
\tag{PI}
$$

where $\gamma \in (0, 1)$ is a discount factor and $c_i \in \mathbb{R}$ represent the $i$-th reward specification. It is important to contrast the formulation in (PI) with the unconstrained, regularized problem commonly found in the literature [4, 19, 20]

$$
\underset{\pi \in \mathcal{P}(\mathcal{S})}{\text{maximize}} \quad V_0(\pi) + \sum_{i=1}^{m} w_i \left(V_i(\pi) - c_i\right),
\tag{$\tilde{\text{PI}}$}
$$

where $w_i \geq 0$ are the regularization parameters. First, (PI) precludes the manual balancing of different requirements through the choice of $w_i$. Even with expert knowledge, tuning these parameters can be as hard as solving the RL problem itself, since there is no straightforward relation between the value of $w_i$ and the value $V_i(\pi^\star)$ given by the final policy. What is more, note that the objective of $(\tilde{\text{PI}})$ can be written as a single value function $\bar{V}(\pi) \triangleq \mathbb{E}_{s,\pi}\left[\sum_{t=0}^{\infty} \gamma^t \bar{r}(s_t, \pi(s_t))\right]$ for $\bar{r}(s_t, \pi(s_t)) = r_0(s_t, \pi(s_t)) + \sum_{i=1}^{m} w_i r_i(s_t, \pi(s_t))$. In other words, choosing the value of $w_i$ amounts to designing a reward that simultaneously encodes different, possibly conflicting, objectives and/or requirements. Given the challenge that can be designing good reward functions for a single task, it is ready that this regularized approach is neither efficient nor effective.

Though promising, solving the constrained RL problem in (PI) is intricate. Indeed, it is both infinite dimensional and non-convex, so that it is in general not tractable in the primal domain. Its dual problem, on the other hand, is convex and has dimensionality equal to the number of constraints. However, since (PI) is not a convex program, its dual problem in general only provides an upper bound on $P^\star$. How good the policy obtained by solving the dual problem is depends on the tightness of this bound. What is more, formulating the problem in the dual domain is at least as hard as solving $(\tilde{\text{PI}})$, which is also infinite dimensional and non-convex. In the sequel, we address these two issues by first showing that (PI) has no duality gap (Section 3), i.e., that the upper bound on $P^\star$ from the dual problem is tight. This implies that (PI) can be solved exactly in the dual domain. Then, we show that we lose (almost) nothing by parametrizing the policies $\pi$ (Section 4), which immediately addresses the issue of dimensionality in (PI)–$(\tilde{\text{PI}})$. Finally, we put forward and analyze a primal-dual algorithm for constrained RL (Section 5), showing that under mild conditions it yields a locally optimal, feasible solution of (PI).

## 3 Constrained Reinforcement Learning Has Zero Duality Gap

Let us start by formalizing the concept of dual problem. Let the vector $\lambda \in \mathbb{R}_+^m$ collect the Lagrange multipliers of the constraints of (PI) and define its Lagrangian as

$$\mathcal{L}(\pi, \lambda) \triangleq V_0(\pi) + \sum_{i=1}^{m} \lambda_i \left( V_i(\pi) - c_i \right). \tag{1}$$

The dual function is then the point-wise maximum of (1) with respect to the policy $\pi$, i.e.,

$$d(\lambda) \triangleq \max_{\pi \in \mathcal{P}(\mathcal{S})} \mathcal{L}(\pi, \lambda). \tag{2}$$

The dual function (2) provides an upper bounds on the value of (PI), i.e., $d(\lambda) \geq P^\star$ for all $\lambda \in \mathbb{R}_+^m$ [22, Section 5.1.3]. The tighter the bound, the closer the policy obtained from (2) is to the optimal solution of (PI). Hence, the dual problem is that of finding the tightest of these bounds:

$$D^\star \triangleq \min_{\lambda \in \mathbb{R}_+^m} d(\lambda). \tag{DI}$$

Note that the dual function (2) can be related to the unconstrained, regularized problem $(\tilde{\text{PI}})$ from Section 2 by taking $\lambda_i = w_i$ in (1). Hence, (2) takes on the optimal value of $(\tilde{\text{PI}})$ for all possible regularization parameters. Problem (DI) then finds the best regularized problem, i.e., that whose value is closest to $P^\star$. It turns out, this problem is tractable if $d(\lambda)$ can be evaluated, since (DI) is a convex program (the dual function is the point-wise maximum of a set of linear functions and is therefore convex) [22, Section 3.2.3].

Despite these similarities, (DI) [and consequently $(\tilde{\text{PI}})$] do not necessarily solve the same problem as (PI). In other words, there need not be a relation between the optimal dual variables $\lambda^\star$ from (DI) or the regularization parameters $w_i$ and the specifications $c_i$ of (PI). This depends on the value of the duality gap $\Delta = D^\star - P^\star$. Indeed, if $\Delta$ is small, then so is the suboptimality of the policies obtained from (DI). In the limit case where $\Delta = 0$, problems (PI)–(DI) and $(\tilde{\text{PI}})$ would all be essentially equivalent. Since (PI) is not a convex program, however, this result does not hold immediately. Still, we calim in Theorem 1 that (PI) has zero duality gap under Slater's conditions. Before stating the Theorem we define the perturbation function associated to problem (PI) which is fundamental for the proof of the result and for future reference. For any $\xi \in \mathbb{R}^n$, the perturbation function associated

to (PI) is defined as

$$P(\xi) \triangleq \max_{\pi \in \mathcal{P}(\mathcal{S})} V_0(\pi) \qquad (\text{PI}')$$
$$\text{subject to } V_i(\pi) \geq c_i + \xi_i, \ i = 1 \ldots m.$$

Notice that $P(0) = P^\star$, the optimal value of (PI). We formally state next the conditions under which Problem (PI) has zero duality gap.

**Theorem 1.** *Suppose that $r_i$ is bounded for all $i = 0, \ldots, m$ and that Slater's condition holds for* (PI). *Then, strong duality holds for* (PI), *i.e.,* $P^\star = D^\star$.

*Proof.* This proof relies on a well-known result from perturbation theory connecting strong duality to the convexity of the perturbation function defined in(PI'). We formalize this result next.

**Proposition 1** (Fenchel-Moreau). *If (i) Slater's condition holds for* (PI) *and (ii) its perturbation function $P(\xi)$ is concave, then strong duality holds for* (PI).

*Proof.* See, e.g., [23, Cor. 30.2.2]. $\square$

Condition (i) of Proposition 1 is satisfied by the hypotheses of Theorem 1. It suffices then to show that the perturbation function is concave [(ii)], i.e., that for every $\xi^1, \xi^2 \in \mathbb{R}^m$, and $\mu \in (0, 1)$,

$$P\left[\mu\xi^1 + (1-\mu)\xi^2\right] \geq \mu P\left(\xi^1\right) + (1-\mu)P\left(\xi^2\right). \qquad (3)$$

If for either perturbation $\xi^1$ or $\xi^2$ the problem becomes infeasible then $P(\xi^1) = -\infty$ or $P(\xi^2) = -\infty$ and thus (3) holds trivially. For perturbations that keep the problem feasible, suppose $P(\xi^1)$ and $P(\xi^2)$ are achieved by the policies $\pi_1 \in \mathcal{P}(\mathcal{S})$ and $\pi_2 \in \mathcal{P}(\mathcal{S})$ respectively. Then, $P(\xi^1) = V_0(\pi_1)$ with $V_i(\pi_1) - c_i \geq \xi_i^1$ and $P(\xi^2) = V_0(\pi_2)$ with $V_i(\pi_2) - c_i \geq \xi_i^2$ for $i = 1, \ldots, m$. To establish (3) it suffices to show that for every $\mu \in (0, 1)$ there exists a policy $\pi_\mu$ such that $V_i(\pi_\mu) - c_i \geq \mu\xi_i^1 + (1-\mu)\xi_i^2$ and $V_0(\pi_\mu) = \mu V_0(\pi_1) + (1-\mu)V_0(\pi_2)$. Notice that any policy $\pi_\mu$ satisfying the previous conditions is a feasible policy for the slack $c_i + \mu\xi_i^1 + (1-\mu)\xi_i^2$. Hence, by definition of the perturbed function (PI'), it follows that

$$P\left[\mu\xi^1 + (1-\mu)\xi^2\right] \geq V_0(\pi_\mu) = \mu V_0(\pi_1) + (1-\mu)V_0(\pi_2) = \mu P\left(\xi^1\right) + (1-\mu)P\left(\xi^2\right). \quad (4)$$

If such policy exists, the previous equation implies (3). Thus, to complete the proof of the result we need to establish its existence. To do so we start by formulating a linear program equivalent to (PI'). Notice that for any $i = 0, \ldots, m$ we can write

$$V_i(\pi) = \int_{(\mathcal{S}\times\mathcal{A})^\infty} \left(\sum_{t=0}^\infty \gamma^t r_i(s_t, a_t)\right) p_\pi(s_0, a_0, \ldots) \, ds_0 \ldots da_0 \ldots. \qquad (5)$$

Since the reward functions are bounded the Dominated Convergence Theorem holds. This allows us to exchange the order of the sum and the integral. Moreover, using conditional probabilities and the Markov property of the transition of the system we can write $V_i(\pi)$ as

$$V_i(\pi) = \sum_{t=0}^\infty \gamma^t \int_{(\mathcal{S}\times\mathcal{A})^\infty} r_i(s_t, a_t) \prod_{u=1}^\infty p(s_u|s_{u-1}, a_{u-1})\pi(a_u|s_u)p(s_0)\pi(a_0|s_0) \, ds_0 \ldots da_0 \ldots. \qquad (6)$$

Notice that for every $u > t$ the integrals with respect to $a_u$ and $s_u$ yield one, since they are integrating density functions. Thus, the previous expression reduces to

$$V_i(\pi) = \sum_{t=0}^\infty \gamma^t \int_{(\mathcal{S}\times\mathcal{A})^t} r_i(s_t, a_t) \prod_{u=1}^t p(s_u|s_{u-1}, a_{u-1})\pi(a_u|s_u)p(s_0)\pi(a_0|s_0) \, ds_0 \ldots ds_t da_0 \ldots da_t. \qquad (7)$$

Notice that the probability density of being at state $s$ and choosing action $a$ under the policy $\pi$ at time $t$ can be written as

$$p_\pi^t(s_t, a_t) = \int_{(\mathcal{S}\times\mathcal{A})^{t-1}} \prod_{u=1}^t p(s_u|s_{u-1}, a_{u-1})\pi(a_u|s_u)p(s_0)\pi(a_0|s_0) \, ds_0 \ldots ds_{t-1} da_0 \ldots da_{t-1}. \qquad (8)$$

Thus, using again the Dominated Convergence Theorem, one can write compactly (7) as

$$V_i(\pi) = \int_{\mathcal{S} \times \mathcal{A}} r_i(s, a) \sum_{t=0}^{\infty} \gamma^t p_\pi^t(s, a) \, ds da. \tag{9}$$

By defining the occupation measure $\rho(s, a) = (1 - \gamma) \sum_{t=0}^{\infty} \gamma^t p_\pi^t(s, a)$ it follows that $(1 - \gamma)V_i(\pi) = \int_{\mathcal{S} \times \mathcal{A}} r_i(s, a)\rho(s, a) \, ds da$. Denote by $\mathcal{M}(\mathcal{S}, \mathcal{A})$ the measures over $\mathcal{S} \times \mathcal{A}$ and define the set $\mathcal{R}$ as the set of all occupation measures induced by the policies $\pi \in \mathcal{P}(\mathcal{S})$ as

$$\mathcal{R} := \left\{ \rho \in \mathcal{M}(\mathcal{S}, \mathcal{A}) \big| \rho(s, a) = (1 - \gamma) \left( \sum_{t=0}^{\infty} \gamma^t p_\pi(s_t = s, a_t = a) \right) \right\}, \tag{10}$$

where It follows from [24, Theorem 3.1] that the set of occupation measures $\mathcal{R}$ is convex and compact. Hence, we can write the following linear program equivalent to (PI′)

$$P(\xi) \triangleq \max_{\rho \in \mathcal{R}} \frac{1}{1 - \gamma} \int_{\mathcal{S} \times \mathcal{A}} r_0(s, a)\rho(s, a) \, ds da$$

$$\text{subject to} \frac{1}{1 - \gamma} \int_{\mathcal{S} \times \mathcal{A}} r_i(s, a)\rho(s, a) \, ds da \geq c_i + \xi_i, \ i = 1, \ldots, m. \tag{PI″}$$

Let $\rho_1, \rho_2 \in \mathcal{R}$ be the occupation measures associated to $\pi_1$ and $\pi_2$. Since, $\mathcal{R}$ is convex, there exists a policy $\pi_\mu \in \mathcal{P}(\mathcal{S})$ such that its corresponding occupation measure is $\rho_\mu = \mu\rho_1 + (1 - \mu)\rho_2 \in \mathcal{R}$. Notice that $\rho_\mu$ satisfies the constraints with slack $c_i + \mu\xi_i^1 + (1 - \mu)\xi_i^2$ for $i = 1, \ldots, m$ since the integral is linear and $\rho_1$ and $\rho_2$ satisfy the constraints with slacks $c_i + \xi_i^1$ and $c_i + \xi_i^2$ respectively. Thus, it follows that

$$P(\mu\xi^1 + (1 - \mu)\xi^2) \geq \frac{1}{1 - \gamma} \int_{\mathcal{S} \times \mathcal{A}} r_0(s, a)\rho_\mu(s, a) \, ds da = \mu V_0(\pi_1) + (1 - \mu)V_0(\pi_2), \tag{11}$$

where we have used again the linearity of the integral. Since $\pi_i$ are such that $V_0(\pi_1) = P(\xi^1)$ and $V_0(\pi_2) = P(\xi^2)$, inequality (3) follows. This completes the proof that the perturbation function is concave. □

Theorem 1 establishes a fundamental equivalence between the constrained (PI) and the dual problem (DI) [and therefore also (P̃I)]. Indeed, since (PI) has no duality gap, its solution can be obtained by solving (DI). What is more, the trade-offs expressed by the $w_i$ in (P̃I) are the same as those expressed by the specifications $c_i$ in the sense that they trace the same Pareto front. Nevertheless, note that the relationship between $c_i$ and $w_i$ is not trivial and that specifying the constrained problem is often considerably simpler. Theorem 1 establishes that this is indeed a valid transformation, since both problems are equivalent. Observe that due to the non-convexity of the objective in RL problems, this result is in fact not immediate.

The theoretical importance of the previous result notwithstanding, it does not yield a procedure to solve (PI) since evaluating the dual function involves a maximization problem that is intractable for general classes of distributions. In the next section, we study the effect of using a finite parametrization for the policies and show that the price to pay in terms of duality gap depends on how "good" the parametrization is. If we consider, for instance, a neural network—which are universal function approximators [12, 25–28]—the loss in optimality can be made arbitrarily small.

## 4 There is (almost) no price to pay by parametrizing the policies

We consider next the problem where the policies are parametrized by a vector $\theta \in \mathbb{R}^p$. This vector could be for instance the coefficients of a neural network or the weights of a linear combination of functions. In this work, we focus our attention however on a widely used class of parametrizations that we term *near-universal*, which are able to model any function in $\mathcal{P}(\mathcal{S})$ to within a stated accuracy. We formalize this concept in the following definition.

**Definition 1.** *A parametrization $\pi_\theta$ is an $\epsilon$-universal parametrization of functions in $\mathcal{P}(\mathcal{S})$ if, for some $\epsilon > 0$, there exists for any $\pi \in \mathcal{P}(\mathcal{S})$ a parameter $\theta \in \mathbb{R}^p$ such that*

$$\max_{s \in \mathcal{S}} \int_{\mathcal{A}} |\pi(a|s) - \pi_\theta(a|s)| \, da \leq \epsilon. \tag{12}$$

The previous definition includes all parametrizations that induce distributions that are close to distributions in $\mathcal{P}(\mathcal{S})$ in total variational norm. Notice that this is a milder requirement than approximation in uniform norm which is a property that has been established to be satisfied by radial basis functions networks [29], reproducing kernel Hilbert spaces [30] and deep neural networks [12]. Notice that the objective function and the constraints in Problem (PI) involve an infinite horizon and thus, the policy is applied an infinite number of times. Hence, the error introduced by the parametrization could a priori accumulate and induce distributions over trajectories that differ considerably from the distributions induced by policies in $\mathcal{P}(\mathcal{S})$. We claim in the following lemma that this is not the case.

**Lemma 1.** *Let $\rho$ and $\rho_\theta$ be occupation measures induced by the policies $\pi \in \mathcal{P}(\mathcal{S})$ and $\pi_\theta$ respectively, where $\pi_\theta$ is an $\epsilon$- parametrization of $\pi$. Then, it follows that*

$$\int_{\mathcal{S} \times \mathcal{A}} |\rho(s,a) - \rho_\theta(s,a)| \, ds da \leq \frac{\epsilon}{1-\gamma}. \tag{13}$$

The previous result, although derived as a technical result required to bound the duality gap for parametric problems, has a natural interpretation. The larger $\gamma$ —the more the operation is concerned about rewards far in the future —the larger the error in the approximation of the occupation measure. Having defined the concept of universal approximator, we shift focus to writing the parametric version of the constrained reinforcement learning problem. This is, to find the parameters that solve (PI), where now the policies are restricted to the functions induced by the chosen parametrization

$$
\begin{aligned}
P_\theta^\star \triangleq \max_\theta \quad & V_0(\theta) \triangleq \mathbb{E}_{s,\pi_\theta} \left[ \sum_{t=0}^\infty \gamma^t r_0(s_t, \pi_\theta(s_t)) \right] \\
\text{subject to} \quad & V_i(\theta) \triangleq \mathbb{E}_{s,\pi_\theta} \left[ \sum_{t=0}^\infty \gamma^t r_i(s_t, \pi_\theta(s_t)) \right] \geq c_i, \ i = 1 \dots m.
\end{aligned} \tag{PII}
$$

Notice that the problem (PII) is similar to the original problem (PI), with the only difference that the expectations are now with respect to distributions induced by the parameter vector $\theta$. As done in the previous section, let $\lambda \in \mathbb{R}^m$ and define the dual function associated to (PII) as

$$d_\theta(\lambda) \triangleq \min_{\theta \in \mathbb{R}^p} \mathcal{L}_\theta(\theta, \lambda) \triangleq \min_{\theta \in \mathbb{R}^p} V_0(\theta) + \sum_{i=1}^m \lambda_i \left( V_i(\theta) - c_i \right), \tag{14}$$

Likewise we define the dual problem as finding the tightest upper bound for (PII)

$$D_\theta^\star \triangleq \underset{\lambda \in \mathbb{R}_+^m}{\text{minimize}} \quad d_\theta(\lambda). \tag{DII}$$

As previously stated, the reason for introducing the parametrization is to turn the original functional optimization problem into a tractable problem in which the optimization variable is a finite dimensional vector of parameters. Yet, there is a cost for introducing the aforementioned parametrization: the duality gap is no longer null. The latter means that the solution obtained through the dual problem is sub-optimal. We claim however that this gap is bounded by a function that is linear with the approximation error $\epsilon$, and thus if the parametrization has a good representation power the price to pay is almost zero. This is the subject of the following theorem.

**Theorem 2.** *Suppose that $r_i$ is bounded for all $i = 0, \dots, m$ by constants $B_{r_i} > 0$ and define $B_r = \max_{i=1 \dots m} B_{r_i}$. Let $\lambda_\epsilon^\star$ be the solution to the dual problem associated to (PI') for perturbation $\xi_i = B_r \epsilon / (1-\gamma)$ for all $i = 1, \dots, m$. Then, under the hypothesis of Theorem 1 it follows that*

$$P^\star \geq D_\theta^\star \geq P^\star - \left( B_{r_0} + \|\lambda_\epsilon^\star\|_1 B_r \right) \frac{\epsilon}{1-\gamma}, \tag{15}$$

*where $P^\star$ is the optimal value of (PI), and $D_\theta^\star$ the value of the parametrized dual problem (DII).*

The implication of the previous result is that there is almost no price to pay by introducing a parametrization. By solving the dual problem (DII) the sub-optimality achieved is of order $\epsilon$, i.e., the error on the representation of the policies. Notice that this error could be made arbitrarily small by increasing the representation ability of the parametrization, by for instance increasing the dimension of the vector of parameters $\theta$. The latter means that if we can compute the dual function it is

possible to solve (PI) approximately. Moreover, working on the dual domain provides two computational advantages; on one hand, the dimension of the problem is the number of constraints in (PI). In addition, the dual function is always convex, hence gradient descent on the dual domain solves the problem of interest. In the next section we propose an algorithm to solve (PI) approximately based on the previous discussion.

Before doing so notice that we have not assumed anything about the feasibility of problem (PII). Notice that if the problem is infeasible then we have that $D_\theta^\star = -\infty$ and thus the upper bound on (15) holds trivially. On the other hand if the problem is infeasible it also means that there is no policy $\pi \in \mathcal{P}(\mathcal{S})$ that satisfies the constraints of (PI) with slack $B_r \epsilon/(1-\gamma)$ since $\theta$ is an $\epsilon$-universal approximation of $\mathcal{P}(\mathcal{S})$. Hence the perturbed problem is infeasible which yields a dual multiplier $\lambda_\epsilon^\star$ that has infinite norm. Thus the right hand side of (15) holds as well. In that sense, as long as the parameterization introduced keeps the problem feasible the price to pay for parameterizing is almost zero.

## 5    Solving Constrained Reinforcement Learning Problems

As previously stated, the dual function is always a convex function since it is the point-wise maximum of linear functions. Thus the dual problem (DII) can be efficiently solved using (sub)gradient descent, with the caveat that because we require the dual iterates to remain in the positive orthant, we include a projection onto this space after taking the gradient step

$$\lambda_{k+1} = [\lambda_k - \eta \partial d_\theta(\lambda_k)]_+ , \tag{16}$$

where $\eta > 0$ is the step-size of the algorithm, $[\cdot]_+$ denotes the projection onto $\mathbb{R}_+^m$ and $\partial d_\theta(\lambda)$ denotes —with a slight abuse of notation —a vector in the subgradient of $d_\theta(\lambda)$. The latter can be computed by virtue of Dankin's Theorem (see e.g. [31, Chapter 3]) by evaluating the constraints in the original problem (PII) at the primal maximizer of the Lagrangian. Thus, the main theoretical difficulty in this computation lies on finding said maximizer since the Lagrangian is non-convex with respect to $\theta$. However, maximizing the Lagrangian with respect to $\theta$ corresponds to learning a policy that uses as reward the following linear combination of rewards

$$r_\lambda(s, a) = r_0(s, a) + \sum_{i=1}^m \lambda_i r_i(s, a). \tag{17}$$

Indeed, using the linearity of the expectation, the cumulative discounted cost for the reward $r_\lambda(s, a)$ yields

$$\mathbb{E}_{s,\pi} \left[ \sum_{t=0}^\infty \gamma^t r_\lambda(s_t, a_t) \right] = \mathbb{E}_{s,\pi_\theta} \left[ \sum_{t=0}^\infty \gamma^t r_0(s_t, a_t) \right] + \sum_{i=1}^m \lambda_i \mathbb{E}_{s,\pi_\theta} \left[ \sum_{t=0}^\infty \gamma^t r_i(s_t, a_t) \right] = \mathcal{L}(\theta, \lambda). \tag{18}$$

And therefore reinforcement learning algorithms such as policy gradient [32] or actor-critic methods [33] can be used to find the parameters $\theta$ such that they maximize the Lagrangian. The good performance of these algorithms is rooted in the fact that they are able to maximize the expected cumulative reward or at least to achieve a value that is close to the maximum. The next assumption formalizes this idea.

**Assumption 1.** *Let $\pi_\theta$ be a parametrization of functions in $\mathcal{P}(\mathcal{S})$ and let $\mathcal{L}_\theta(\theta, \lambda)$ with $\lambda \in \mathbb{R}_+^m$ be the Lagrangian associated to* (PII). *Denote by $\theta^\star(\lambda), \theta^\dagger(\lambda) \in \mathbb{R}^P$ the maximum of $\mathcal{L}(\theta, \lambda)$ and a local maximum respectively achieved by a generic reinforcement learning algorithm. Then, there exists $\delta > 0$ such that for all $\lambda \in \mathbb{R}_+^m$ it holds that $\mathcal{L}_\theta(\theta^\star(\lambda), \lambda) \leq \mathcal{L}_\theta(\theta^\dagger(\lambda), \lambda) + \delta$.*

Notice that the previous assumption only means that we are able to solve the regularized unconstrained problem approximately. This means that the parameter at time $k + 1$ is

$$\theta_{k+1} \approx \underset{\theta \in \mathbb{R}^p}{\operatorname{argmax}} \, \mathcal{L}(\lambda_k, \theta). \tag{19}$$

Then, the dual variable is updated following the gradient descent scheme suggested in (16), where we replace the subgradient of the dual function by the constraint of the primal problem (PII). Defining $\hat{\partial} d_k \triangleq V(\theta_{k+1}) - s$, the update yields

$$\lambda_{k+1} = \left[ \lambda_k - \eta \hat{\partial} d_k \right]_+ = \left[ \lambda_k - \eta \left( V(\theta_{k+1}) - s \right) \right]_+ . \tag{20}$$

**Algorithm 1** dualDescent

---

**Input:** $\eta$
1: *Initialize*: $\theta_0 = 0$, $\lambda_0 = 0$
2: **for** $k = 0, 1 \dots$
3:    Compute an approximation of $\theta_{k+1} \approx \operatorname{argmax} \mathcal{L}_\theta(\theta, \lambda_k)$ with a RL algorithm
4:    Compute the dual ascent step $\lambda_{k+1} = [\lambda_k - \eta (V(\theta_{k+1}) - s)]_+$.
5: **end**

---

The algorithm given by (19)–(20) is summarized under Algorithm 1. The previous algorithm relies on the fact that the $\hat{\partial} d_k$ does not differ much from $\partial d_\theta(\lambda_k)$. We claim in the following proposition that this is the case. In particular, we establish that the constraint evaluation does not differ from the subgradient in more than $\delta$, the error on the primal maximization defined in Assumption 1.

**Proposition 2.** *Under Assumption 1, the constraint in* (PII) *evaluated at a local maximizer of Lagrangian* $\theta^\dagger(\lambda)$ *approximate the subgradient of the dual function* (14). *In particular it follows that*

$$d_\theta(\lambda) - d_\theta(\lambda_\theta^\star) \leq (\lambda - \lambda_\theta^\star)^\top \left( V(\theta^\dagger(\lambda)) - s \right) + \delta. \tag{21}$$

The previous proposition is key in establishing convergence of the algorithm proposed since allows us to claim that the dual updated is an approximation of a dual descent step. We formalize this result next and we establish a maximum number of dual steps required to achieve a desired accuracy.

**Theorem 3.** *Let* $\pi_\theta$ *be an* $\epsilon$ *universal parametrization of* $\mathcal{P}(\mathcal{S})$ *according to Definition 1,* $B_r = \max_{i=1\dots m} B_{r_i}$ *with* $B_{r_i} > 0$ *bounds on the rewards* $r_i$ *and* $\gamma \in (0, 1)$ *be the discount factor. Then, if Slater's conditions hold for* (PII), *under Assumption 1 and for any* $\varepsilon > 0$, *the sequence of updates of Algorithm 1 with step size* $\eta$ *converges in* $K > 0$ *steps, with*

$$K \leq \frac{\|\lambda_0 - \lambda_\theta^\star\|^2}{2\eta\varepsilon}, \tag{22}$$

*to a neighborhood of* $P^\star$ *–the solution of* (PI)– *satisfying*

$$P^\star - (B_{r_0} + \|\lambda_\epsilon^\star\|_1 B_r) \frac{\epsilon}{1 - \gamma} \leq d_\theta(\lambda_K) \leq P^\star + \eta \frac{B}{2} + \delta + \varepsilon. \tag{23}$$

*where* $B = \sum_{i=1}^m \left( B_{r_i}/(1 - \gamma) - c_i \right)^2$ *and* $\lambda^\star$ *is the solution of* (DI).

The previous result establishes a bound on the number of dual iterations required to converge to a neighborhood of the optimal solution. This bound is linear with the inverse of the desired accuracy $\varepsilon$. Notice that the size of the neighborhood to which the dual descent algorithm converges depends on the representation ability of the parametrization chosen, and the goodness of the solution of the maximization of the Lagrangian. Since the cost of running policy gradient or actor-critic algorithms until convergence before updating the dual variable might result in an algorithm that is computationally prohibitive, an alternative that is common in the context of optimization is to update both variables in parallel [34]. This idea can be applied in the context of reinforcement learning as well, where a policy gradient —or actor critic as in [7, 11] —update is followed by an update of the multipliers along the direction of the constraint violation. In these algorithms the update on the policy is on a faster scale than the update of the multipliers, and therefore they operate from a theoretical point of view as (1). In particular, the proofs in [7, 11] rely on the fact that this different time-scale is such that allows to consider the multiplier as constant.

## 6   Numerical Example

In this section, we include a numerical example in order to showcase the consequences of our theoretical results. As an illustrative example, we consider a gridworld navigation scenario. This scenario, illustrated in Figure 1, consists of an agent attempting to navigate from a starting position to a goal. To do so, the agent must cross from the left side of the world to the right side using either one of two bridges. The bridge above is deemed "unsafe". The agents uses a softmax policy with four possible actions (moving up, `down`, `left`, and `right`) over a table-lookup of states and actions. The agent receives a reward $r(s, a) = 10$ for reaching the goal and a reward of $r(s, a) = -1$ for

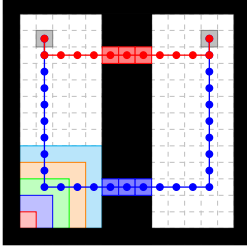

Figure 1: Safe (blue) and unsafe (red) optimal path. Parametrization coarseness is on the bottom left.

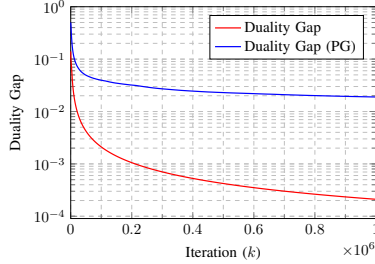

Figure 2: Duality gap of the policies.

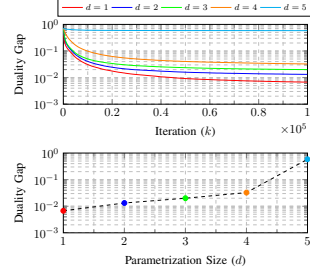

Figure 3: Effect of parametrization coarseness.

each step it wanders outside of goal. The scenario is designed such that the shortest path requires crossing the unsafe path (red bridge), while the safe path (blue bridge) requires a longer detour. Using our formulation, we constrain the agent to not cross the unsafe bridge with $99\%$ probability.

We train the agent via Algorithm 1, agent and plot in Fig. 2, the resulting normalized duality gap. We consider two cases, an inexact primal maximization via policy gradient and, an exact primal maximization. In order to obtain the global primal minimizer, for a given value of the dual variables $\lambda$, the optimal primal minimizer can be easily found via Dijkstra's algorithm. We show that by solving Step 4 of Algorithm 1 exactly the duality gap effectively vanishes (red curve). We also showcase a curve in which Step 4 is replaced by a single policy gradient step (blue curve). Since the minimization in Step 4 is done approximately, the duality gap decreases at a slower rate and will only converge to a neighborhood of zero (as per Theorem 3). In any of the two cases, ultimately, the agent learns to navigate from start to goal by crossing the safe bridge (blue path in Fig. 1).

Now, we turn our attention to the effect of the parametrization size. We consider parametrization of different coarseness via state aggregation, as shown in Fig. 1. This will correspond, as per Definition 1, in parametrizations with lager values of $\epsilon$, i.e., looser approximators. Figure 3 displays the effect of using coarser parametrizations, as the parametrization becomes coarser, the duality gap increases (as per Theorem 2). Specially, for very coarse parametrizations (such as the cyan case), the agent cannot learn a successful policy due to the poor covering properties of its parametrization and resultantly such problem will have a large duality gap.

## 7 Discussion

Throughout this work we have developed a duality theory for constrained reinforcement learning problems. In particular we have established that for policies belonging to a general class of distributions, the duality gap of this problems is null and therefore by solving the problem on the dual domain —which always yields a finite dimensional convex problem —yields the same result as solving the original problem directly. Moreover, it establishes the equivalence between the constrained problem and the regularized problem —or manual selection of multipliers —in the sense that both problems track the same Pareto optimal front.

These theoretical implications however do not imply that it is always possible to solve the problem. To be able to solve the dual problem, one is required to evaluate the dual function, which might result intractable in several problems, for instance in cases where arbitrary policies are considered. To overcome this limitation, we have shown that for sufficiently rich parametrizations the zero duality gap result holds approximately. However, for the most part, the parametrizations considered in the literature are not necessarily universal approximators of distributions since in general the output of the neural network reduces to the mean —and in some cases the variance —of a distribution.

Regardless of these limitations, the primal dual algorithm considered here and those proposed in [7, 11] provide a manner to solve constrained policy optimization problems without the need to perform an exhaustive search over the weights that we assign to each reward function, as it is the case in [4, 19, 20]. Likewise, the need of imposing constraints might arise directly from the algorithm design, this is for instance the case in Trust Region Policy Optimization [35], where a constraint on the divergence of the policy is included. Although our theorems do not guarantee that the zero duality gap result holds under these constraints, since they reduce to a projection onto a convex set it would not be surprising that it could be adapted.

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
