[Supplementary Material]

# A Proofs

*Proof of Lemma 1.* Let us start by writing the left hand side of (13) as

$$\int_{\mathcal{S}\times\mathcal{A}}|\rho(s,a)-\rho_\theta(s,a)|\,dsda = (1-\gamma)\int_{\mathcal{S}\times\mathcal{A}}\left|\sum_{t=0}^{\infty}\gamma^t\left(p_\pi^t(s,a)-p_\theta^t(s,a)\right)\right|dsda \qquad (24)$$

Using the triangle inequality, we upper bound the previous expression as

$$\int_{\mathcal{S}\times\mathcal{A}}|\rho(s,a)-\rho_\theta(s,a)|\,dsda \leq (1-\gamma)\sum_{t=0}^{\infty}\gamma^t\int_{\mathcal{S}\times\mathcal{A}}\left|p_\pi^t(s,a)-p_\theta^t(s,a)\right|dsda. \qquad (25)$$

Notice that to complete the proof it suffices to show that the right hand side of the previous expression is bounded by $\epsilon/(1-\gamma)$. We next work towards that end, and we start by bounding the difference $|p_\pi^t(s,a)-p_\theta^t(s,a)|$. Notice that this difference can be upper bounded using the triangle inequality as

$$\left|p_\pi^t(s,a)-p_\theta^t(s,a)\right| \leq p_\pi^t(s)\left|\pi(a|s)-\pi_\theta(a|s)\right| + \pi_\theta(a|s)\left|p_\pi^t(s)-p_\theta^t(s)\right|. \qquad (26)$$

Since $\pi_\theta$ is an $\epsilon$-approximation of $\pi$, it follows from Definition 1 that

$$\int_{\mathcal{S}\times\mathcal{A}}p_\pi^t(s)\left|\pi(a|s)-\pi_\theta(a|s)\right|\,dsda \leq \epsilon\int_{\mathcal{S}}p_\pi^t(s)\,ds = \epsilon, \qquad (27)$$

where the last equality follows from the fact that $p_\pi^t(s)$ is a density. We next work towards bounding the integral of the second term in (26). Using the fact that $\pi_\theta(a|s)$ is a density, it follows that

$$\int_{\mathcal{S}\times\mathcal{A}}\pi_\theta(a|s)\left|p_\pi^t(s)-p_\theta^t(s)\right|\,dsda = \int_{\mathcal{S}}\left|p_\pi^t(s)-p_\theta^t(s)\right|\,ds. \qquad (28)$$

Notice that the previous difference is zero for $t=0$ and for any $t>0$ it can be upper bounded by

$$\int_{\mathcal{S}}\left|p_\pi^t(s)-p_\theta^t(s)\right|\,ds \leq \int_{\mathcal{S}}\int_{\mathcal{S}\times\mathcal{A}}p(s|s',a')\left|p_\pi^{t-1}(s',a')-p_\theta^{t-1}(s',a')\right|\,dsds'da'$$

$$= \int_{\mathcal{S}\times\mathcal{A}}\left|p_\pi^{t-1}(s',a')-p_\theta^{t-1}(s',a')\right|\,ds'da' \qquad (29)$$

Combining the bounds derived in (25), (27), (29) we have that

$$(1-\gamma)\sum_{t=0}^{\infty}\gamma^t\int_{\mathcal{S}\times\mathcal{A}}\left|\left(p_\pi^t(s,a)-p_\theta^t(s,a)\right)\right|\,dsda \leq$$

$$(1-\gamma)\sum_{t=0}^{\infty}\gamma^t\epsilon + (1-\gamma)\sum_{t=1}^{\infty}\gamma^t\int_{\mathcal{S}\times\mathcal{A}}\left|\left(p_\pi^{t-1}(s,a)-p_\theta^{t-1}(s,a)\right)\right|\,dsda. \qquad (30)$$

Notice that the first term on the right hand side of the previous expression is the sum of the geometric multiplied by $1-\gamma$. Hence we have that $(1-\gamma)\sum_{t=0}^{\infty}\gamma^t\epsilon = \epsilon$. The second term on the right hand side of the previous expression is in fact the same as the term on the left hand side of the expression multiplied by the discount factor $\gamma$. Thus, rearranging the terms, the previous expression implies that

$$(1-\gamma)\sum_{t=0}^{\infty}\gamma^t\int_{\mathcal{S}\times\mathcal{A}}\left|\left(p_\pi^t(s,a)-p_\theta^t(s,a)\right)\right|\,dsda \leq \frac{\epsilon}{1-\gamma}. \qquad (31)$$

This completes the proof of the Lemma. $\qquad\square$

*Proof of Theorem 2.* Notice that the dual functions $d(\lambda)$ and $d_\theta(\lambda)$ associated to the problems (PI) and (PII) respectively are such that for every $\lambda$ we have that $d_\theta(\lambda) \leq d(\lambda)$. The latter follows from the fact that the set of maximizers of the Lagrangian for the parametrized policies is contained in the set of maximizers of the non-parametrized policies. In particular, this holds for $\lambda^\star$ the solution of the dual problem associated to (PI). Hence we have the following sequence of inequalities

$$D^\star = d(\lambda^\star) \geq d_\theta(\lambda^\star) \geq D_\theta^\star, \qquad (32)$$

where the last inequality follows from the fact that $D_\theta^\star$ is the minimum of (DII). The zero duality gap established in Theorem 1 completes the proof of the upper bound for $D_\theta^\star$. We next work towards proving the lower bound for $D_\theta^\star$. Let us next write the dual function of the parametrized problem (DII) as

$$d_\theta(\lambda) = d(\lambda) - \left( \max_{\pi \in \mathcal{P}(\mathcal{S})} \mathcal{L}(\pi, \lambda) - \max_{\theta \in \mathbb{R}^p} \mathcal{L}_\theta(\theta, \lambda) \right) \tag{33}$$

Let $\pi^\star \triangleq \mathrm{argmax}_{\pi \in \mathcal{P}(\mathcal{S})} \mathcal{L}(\pi, \lambda)$ and let $\theta^\star$ be an $\epsilon$-approximation of $\pi^\star$. Then, by definition of the maximum it follows that

$$d_\theta(\lambda) \geq d(\lambda) - (\mathcal{L}(\pi^\star, \lambda) - \mathcal{L}_\theta(\theta^\star, \lambda)) \tag{34}$$

We next work towards a bound for $\mathcal{L}(\pi^\star, \lambda) - \mathcal{L}_\theta(\theta^\star, \lambda)$. To do so, notice that we can write the difference in terms of the occupation measures where $\rho^\star$ and $\rho_\theta^\star$ are the occupation measures associated to the the policies $\pi^\star$ and the policy $\pi_{\theta^\star}$

$$\mathcal{L}(\pi^\star, \lambda) - \mathcal{L}_\theta(\theta^\star, \lambda) = \int_{\mathcal{S} \times \mathcal{A}} \left( r_0 + \lambda^\top r \right) \left( d\rho^\star(\lambda) - d\rho_\theta^\star(\lambda) \right). \tag{35}$$

Since $\pi_{\theta^\star}$ is by definition an $\epsilon$ approximation of $\pi^\star$ it follows from Lemma 1 that

$$\int_{\mathcal{S} \times \mathcal{A}} |d\rho^\star(\lambda) - d\rho_\theta^\star(\lambda)| \leq \frac{\epsilon}{1 - \gamma}. \tag{36}$$

Using the bounds on the the reward functions we can upper bound the difference $\mathcal{L}(\pi^\star, \lambda) - \mathcal{L}_\theta(\theta^\star, \lambda)$ by

$$\mathcal{L}(\pi^\star, \lambda) - \mathcal{L}_\theta(\theta^\star, \lambda) \leq (B_{r_0} + \|\lambda\|_1 B_r) \frac{\epsilon}{1 - \gamma}. \tag{37}$$

Combining the previous bound with (34) we can lower bound $d_\theta(\lambda)$ as

$$d_\theta(\lambda) \geq d(\lambda) - (B_{r_0} + \|\lambda\|_1 B_r) \frac{\epsilon}{1 - \gamma} \tag{38}$$

Let us next define $d_\epsilon(\lambda) = d(\lambda) - B_r \epsilon / (1 - \gamma) \|\lambda\|_1$, and notice that in fact $d_\epsilon(\lambda)$ is the dual function associated to Problem (PI$'$) with $\xi_i = B_r \epsilon / (1 - \gamma)$ for all $i = 1, \ldots, m$. With this definition, (38) reduces to

$$d_\theta(\lambda) \geq d_\epsilon(\lambda) - B_{r_0} \frac{\epsilon}{1 - \gamma}. \tag{39}$$

Since the previous expression holds for every $\lambda$, in particular it holds for $\lambda_\theta^\star$, the dual solution of the parametrized problem (DII). Thus, we have that

$$D_\theta^\star \geq d_\epsilon(\lambda_\theta^\star) - B_{r_0} \frac{\epsilon}{1 - \gamma}. \tag{40}$$

Recall that $\lambda_\epsilon^\star = \mathrm{argmin}\, d_\epsilon(\lambda)$, and use the definition of the dual function to lower bound $D_\theta^\star$ by

$$D_\theta^\star \geq \max_{\pi \in \mathcal{P}(\mathcal{S})} V_0(\pi) + \sum_{i=1}^m \lambda_{\epsilon,i}^\star \left( V_i(\pi) - c_i - B_r \frac{\epsilon}{1 - \gamma} \right) - B_{r_0} \frac{\epsilon}{1 - \gamma}. \tag{41}$$

By definition of maximum, we can lower bound the previous expression by substituting by any $\pi \in \mathcal{P}(\mathcal{S})$. In particular, we select $\pi^\star$ the solution to (PI)

$$D_\theta^\star \geq V_0(\pi^\star) + \sum_{i=1}^m \lambda_{\epsilon,i}^\star \left( V_i(\pi^\star) - c_i - B_r \frac{\epsilon}{1 - \gamma} \right) - B_{r_0} \frac{\epsilon}{1 - \gamma}. \tag{42}$$

Since $\pi^\star$ is the optimal solution to (PI) it follows that $V_i(\pi^\star) - c_i \geq 0$ and since $\lambda_{\epsilon,i}^\star \geq 0$ the previous expression reduces to

$$D_\theta^\star \geq V_0(\pi^\star) - (B_{r_0} + B_r \|\lambda_\epsilon^\star\|) \frac{\epsilon}{1 - \gamma} = P^\star - (B_{r_0} + B_r \|\lambda_\epsilon^\star\|) \frac{\epsilon}{1 - \gamma} \tag{43}$$

Which completes the proof of th result $\qquad \square$

*Proof of Proposition 2.* Let $\lambda_\theta^\star$ be the solution of the parametrized dual problem (DII). Then we can write the difference of the dual function evaluated at an arbitrary $\lambda \in \mathbb{R}_+^m$ and $\lambda_\theta^\star$ as

$$d_\theta(\lambda) - d_\theta(\lambda_\theta^\star) = \max_\theta \mathcal{L}_\theta(\theta, \lambda) - \max_\theta \mathcal{L}_\theta(\theta, \lambda_\theta^\star) \leq \mathcal{L}_\theta(\theta^\star(\lambda), \lambda) - \mathcal{L}_\theta(\theta^\dagger(\lambda), \lambda_\theta^\star). \quad (44)$$

It follows from Assumption 1 that there exists $\delta > 0$ such that $\mathcal{L}_\theta(\theta^\star(\lambda), \lambda) \leq \mathcal{L}_\theta(\theta^\dagger(\lambda), \lambda) + \delta$, thus we can upper bound the right hand side of the previous inequality by

$$\mathcal{L}_\theta(\theta^\star(\lambda), \lambda) - \mathcal{L}_\theta(\theta^\dagger(\lambda), \lambda_\theta^\star) \leq \mathcal{L}_\theta(\theta^\dagger(\lambda), \lambda) + \delta - \mathcal{L}(\theta^\dagger(\lambda), \lambda_\theta^\star) = (\lambda - \lambda_\theta^\star)^\top \left(V(\theta^\dagger(\lambda)) - c\right) + \delta. \quad (45)$$

Combining the two upper bounds completes the proof of the proposition. $\qquad \square$

*Proof of Theorem 3.* We start by showing the lower bound, which in fact holds for any $\lambda$. Notice that for any $\lambda$ and by definition of the dual problem it follows that $d_\theta(\lambda) \geq D_\theta^\star$. Combining this bound with the result of Theorem 2 it follows that

$$d_\theta(\lambda) \geq P^\star - \left(B_{r_0} + \|\lambda_\epsilon^\star\|_1 B_r\right) \frac{\epsilon}{1-\gamma}. \quad (46)$$

To show the upper bound we start by writing the difference between the dual multiplier $k+1$ and the solution of (DII) in terms of the iteration at time $k$. Since $\lambda_\theta^\star \in \mathbb{R}_+^m$ and using the non-expansive property of the projection it follows that

$$\left\|\lambda_{k+1} - \lambda_\theta^\star\right\|^2 \leq \left\|\lambda_k - \eta\left(V(\theta^\dagger(\lambda_k)) - c\right) - \lambda_\theta^\star\right\|^2 \quad (47)$$

Expanding the square and using that $B = \sum_{i=1}^m \left(B_{r_i}/(1-\gamma) - c_i\right)^2$ is a bound on the norm squared of $V(\theta) - s$ it follows that

$$\left\|\lambda_{k+1} - \lambda_\theta^\star\right\|^2 \leq \left\|\lambda_k - \lambda_\theta^\star\right\|^2 - 2\eta\left(\lambda_k - \lambda_\theta^\star\right)^\top \left(V(\theta^\dagger(\lambda_k)) - c\right) + \eta^2 B. \quad (48)$$

Using the result of Proposition 2 we can further upper bound the inner product in the previous expression by the difference of the dual function evaluated at $\lambda_k$ and $\lambda_\theta^\star$ plus $\delta$, the error in the solution of the primal maximization,

$$\left\|\lambda_{k+1} - \lambda_\theta^\star\right\|^2 \leq \left\|\lambda_k - \lambda_\theta^\star\right\|^2 + 2\eta\left(\delta + d_\theta(\lambda_\theta^\star) - d_\theta(\lambda_k)\right) + \eta^2 B. \quad (49)$$

Defining $\alpha_k = 2(\delta + d_\theta(\lambda_\theta^\star) - d_\theta(\lambda_k)) + \eta B$ and writing recursively the previous expression yields

$$\left\|\lambda_{k+1} - \lambda_\theta^\star\right\|^2 \leq \left\|\lambda_0 - \lambda_\theta^\star\right\|^2 + \eta \sum_{j=0}^k \alpha_j. \quad (50)$$

Since $d_\theta(\lambda_\theta^\star)$ is the minimum of the dual function, the difference $d_\theta(\lambda_\theta^\star) - d_\theta(\lambda_k)$ is always negative. Thus, when $\lambda_k$ is not close to the solution of the dual problem $\alpha_k$ is negative. The latter implies that the distance between $\lambda_k$ and $\lambda_\theta^\star$ is reduced by virtue of (50). To be formal, for any $\varepsilon > 0$, when $a_j > -2\varepsilon$ we have that

$$d_\theta(\lambda_j) - d_\theta(\lambda_\theta^\star) \leq \eta \frac{B}{2} + \delta + \varepsilon. \quad (51)$$

Using the result of Theorem 2 we can upper bound $D_\theta^\star$ by $P^\star$ which establishes the neighborhood defined in (23). We are left to show that the number of iterations required to do so is bounded by

$$K \leq \frac{\left\|\lambda_0 - \lambda_\theta^\star\right\|^2}{2\eta\varepsilon}. \quad (52)$$

To do so, let $K > 0$ be the first iterate in the neighborhood (23). Formally, $K = \min_{j\in\mathbb{N}} \alpha_j > -2\varepsilon$. Then it follows from the recursion that

$$\left\|\lambda_K - \lambda_\theta^\star\right\|^2 \leq \left\|\lambda_0 - \lambda_\theta^\star\right\|^2 - 2K\eta\varepsilon. \quad (53)$$

Since $\left\|\lambda_K - \lambda_\theta^\star\right\|^2$ is positive the previous expression reduces to $2K\eta\epsilon \leq \left\|\lambda_0 - \lambda_\theta^\star\right\|^2$. Which completes the proof of the result. $\qquad \square$