[Reviews · NeurIPS 2019]

Reviewer 1



The paper studies a form of constrained reinforcement learning in which the constraints are bounds on the value functions for auxiliary rewards. This allows a more expressive formulation than the common approach of defining the reward as a linear combination of multiple objectives. The authors show that under certain conditions, the constraint optimization problem has zero duality gap, implying that a solution can be found by solving the dual optimization problem, which is convex. The authors also extend this analysis to the case for which the policy is parameterized. Theorem 1 assumes that Slater's condition holds, which is problematic for two reasons. Slater's condition is usually defined for convex constraints, but the authors specifically state that the constraints in PI are non-convex. Moreover, according to the common definition, the duality gap is zero when Slater's condition holds, which is precisely what the theorem claims to prove. I think there are important details missing from the assumption that the authors need to spell out explicitly. In fact, I think a more interesting contribution would be to show precisely for which reward functions that Slater's condition holds. As it stands I have poor intuition regarding the types of reward functions that will satisfy Slater's condition. The decision problem PI considers a special case in which constraints are all formulated as a bound on a given value function. What would happen if you allow other types of constraints as well? To what degree does the zero duality gap depend on this specific form of constraint? Top of page 4: "Let [...] P(xi) be the optimal value of ~PI for the perturbation xi" -> should this not be PI rather than ~PI? Same question at the top of page 5: "linear program equivalent to ~PI". I cannot find any mistakes in the derivations of Sections 4 and 5. However, I think it is an exaggeration to say that the parameterization comes at almost no price. First of all, the discount factor is usually close to 1, so the quotient epsilon / ( 1 - gamma ) could grow quite large. Second, although the result holds in theory, it is well known that the bound from the universal approximation theorem is hard to achieve in practice, even if the representation is expressive enough. How is the projection onto R_+^m done? Just taking the maximum of 0 and x? Assumption 1 does not appear realistic to me, for the same reason as outlined above: it is difficult to achieve hard performance bounds using e.g. deep learning. POST-REBUTTAL: I appreciate the authors' effort to clarify the underlying assumptions. The assumption regarding the existence of a feasible policy is still strong, since presumably one is allowed to freely define the reward specifications s_i. However, I agree that it is non-trivial to determine that a zero duality gap holds when the assumption holds. Moreover, the bound on the total variational norm is indeed milder than that for the uniform norm, but the bound is still worst-case over the states. The title of Section 4 gives an impression that in my view is too good to be true. Even the unconstrained RL problem has few theoretical guarantees when combined with function approximation. As a side note, I think your definition of the value function in the derivation of the proof of Theorem 1 (in terms of occupation measures) is similar to that of Lewis and Puterman in "Bias Optimality". Also I just noticed that you use "s_i" both for states and reward specifications which is confusing.

Reviewer 2



The paper looks at reinforcement learning with constraints and provides an analysis of the problem. The authors prove that under common assumptions, the problem has zero duality gap and can be solved exactly in the dual domain. This finding doesn’t necessarily hold when the learned policies are function approximators but the authors provide further arguments to show that under the right type of parameterization, convergence to the optimal solution can still be shown to hold. The paper also describes an algorithm for optimizing the constrained RL objective. I haven’t discovered any gaps, mistakes or other problems in the proofs and argumentation. The paper cites relevant prior work but could be improved by discussing some of this work in more detail. The proposed algorithm differs very little from previous approaches based on primal-dual algorithms and it would have been insightful if there was more discussion about how this algorithm differs and why it would be better or at least complementary. Without this discussion, the presentation of the algorithm only serves as a prototype for which the analysis of the required number of updates is done without getting more insight in the efficiency and expected behavior of similar algorithms that have already been shown to be useful empirically. I find the novelty of some of the individual arguments hard to judge but I do think that the main result is important and that the analyses provide useful insights. I couldn’t find any other work on constrained RL that even mentions the duality gap, so even without the result that the gap doesn’t exist for general policies under certain conditions, the paper points out an important issue that seems to have been glossed over so far. Since the paper mostly shows that the issue is actually not expected to be a problem in practice, the impact is mostly a firmer understanding of why previous approaches work but it also gives some new intuitions about why they might fail if function approximators are not flexible enough. The presented algorithm doesn’t seem to differ much from prior work and without empirical results, it is not possible to judge its value as a contribution on top of its usefulness as an illustrative example. The paper is mostly easy to follow but could be more clear about the main contributions of the paper and how they relate to previous work.

Reviewer 3



This paper proposes a dual approach for the constrained reinforcement learning (CRL) problem. The authors show that the CRL problem has zero duality gap and thus can be solved by solving its dual problem which is convex. The optimality results of parametrized policies and primal-dual algorithms are also given in the paper. The constrained reinforcement learning problem is interesting and important in practice. This paper gives a good theoretical result on this problem. It would be better if the authors could add an experimental section in the paper. UPDATE:After reading the author’s rebuttal, I have chosen to increase my score for the following reason: The authors provide a numerical section in the rebuttal.

[Author Response · NeurIPS 2019]

**Reviewer 1**   We understand the reviewer concerns, but point that the *strong duality of constrained RL is in fact not a trivial result*. Indeed, though *Slater's condition* is often used in the context of convex optimization and does imply strong duality in this specific case, the constrained RL problem in (PI) is not convex, as the reviewer correctly points out. Hence, even if Slater's condition holds, it is not immediate that it has no duality gap. Moreover, what we mean by "Slater's condition" in Theorem 1 (and Proposition 1) is not that the constraints are convex, but simply that *there exists a policy $\pi^\dagger$ (which need not be optimal) that is strictly feasible, i.e., $V_i(\pi^\dagger) > s_i$ for all $i = 1, \ldots m$*. Hence, it is not an assumption on the shape of the reward but on the feasibility of the problem, which in practice is fairly mild: if the RL problem has any solution, even one that is tight, an arbitrarily small relaxation of the constraints will make this solution strictly feasible. Nevertheless, we agree with the reviewer that this and other assumptions in the paper should be better explained. For instance, *Assumption 1* simply makes sure that we are able to find good solutions to unconstrained RL problems, i.e., that the value function achieved by the learned policy is close to the optimal one. In fact, Theorem 3 implies in broad terms that *if we know how to solve unconstrained RL problems well, then we also know how to solve constrained ones*. In the case of the $\epsilon$-*universality* assumption, note that only requires that the parametrization be able to approximate probability distributions "on average" (in TV norm), which is considerably milder than the worst case (uniform) results typically derived in the literature. Finally, the reviewer raises an excellent point that this error leads to an $\mathcal{O}(\epsilon/(1-\gamma))$ duality gap, which may be quite large for $\gamma \approx 1$. This is a fundamental result about RL that characterizes the *trade-off between the quality of the parametrization and the difficulty of the problem*. Indeed, when a lot of weight is placed on the long-term behavior of the agent, good policies must dictate actions with extreme precision as they cannot afford any mistake. Achieving good performance then requires having more precise descriptions of policies. We will clarify and make these points more explicit in the final version of the paper. As for the *constraints being defined in terms of value functions*, though other forms of probabilistic constraints can be used, the stochastic nature of MDPs obstructs the use of Theorem 1 in the presence of deterministic constraints. The reviewer is also correct in that *projections onto $\mathbb{R}_+^m$ involve a simple elementwise maximum*.

**Reviewer 2**   The reviewer is correct that the *main contribution of the work is to provide theoretical guarantees in support of existing approaches* to constrained RL, i.e., our main result is the strong duality of constrained RL for rich parametrizations. In that sense, the algorithm is indeed only described to make the results more concrete, namely, so that we can obtain Theorem 3 and show that if we know how to solve unconstrained RL problems well, we also know how to solve constrained ones. We agree that the remark in contribution 3 in the introduction does not make these points sufficiently clear and we fully intend to rewrite the abstract and introduction to make this explicit in the final version of the paper. We will also develop the discussion of existing methods more deeply in the related work section to *better give credit to existing algorithms*. If the reviewers agree that it would be appropriate to change the title, we propose to name the paper by its main contribution: "*Constrained reinforcement learning has zero duality gap*." If the reviewers have other suggestions, we would welcome their feedback.

**Reviewer 3**   As suggested by the reviewer, we will include a *numerical section* showcasing our results in the final version of the manuscript. Below, we display a few plots to showcase the type of numerical examples we will include. We use a simple navigation scenario for which we can obtain the optimal policy (e.g., using a Dijkstra-type algorithm) so as to illustrate the theoretical results. In this problem (Fig. 1), an agent must cross from one side of the world to another using bridges, one of which is "unsafe" (we constrain the agent to not use this bridge). In Fig. 2, we show that by solving step 4 of Algorithm 1 exactly the duality gap effectively vanishes (red curve). We also showcase a curve in which step 4 is replaced by a single policy gradient step (blue curve). Since the minimization in step 4 is done approximately, the duality gap decreases at a slower rate and will only converge to a neighborhood of zero (Theorem 3). Fig 3 displays the effect of using coarser parametrizations (i.e., larger $\epsilon$ in Definition 1) by forcing the policy to be the same over sets of adjacent states (as illustrated in Fig 1). Note that as the parametrization becomes coarser, the duality gap increases (as per Theorem 2).

Figure 1: Safe (blue) and unsafe (red) optimal path. Coarseness of parametrization is shown on the bottom left.

Figure 2: Gap of optimal policy and policy gradient.

Figure 3: Effect of coarseness of the parametrization

[Meta-Review · NeurIPS 2019]

After discussion, the reviewers converge to an accept decision. Please carefully consider the reviewers' comments in revising your paper.